

**Projected effects of vegetation feedback on drought characteristics of**
**West Africa using a coupled regional land–vegetation–climate model**
[1]Muhammad Shafqat Mehboob, [1]Yeonjoo Kim, [1]Jaehyeong Lee, [2]Myoung-Jin Um, [3]Amir Erfanian,
[4]Guiling Wang
[1]Department of Civil and Environmental Engineering, Yonsei University, Seoul, 03722, South Korea
[2]Department of Civil Engineering, Kyonggi University, Suwon-si Gyeonggi-do, 16227, South Korea
[3]Department of Atmospheric and Oceanic Sciences, University of California, Los Angeles, 90095, USA
[4]Department of Civil and Environmental Engineering, University of Connecticut, Storrs, 06269, USA
**Correspondence***:* Yeonjoo Kim (yeonjoo.kim@yonsei.ac.kr)
**Abstract.** This study investigates the projected effect of vegetation feedback on drought conditions in West Africa using a
regional climate model coupled to the National Center for Atmospheric Research Community Land Model, the carbon-nitrogen
(CN) module, and the dynamic vegetation (DV) module (RegCM-CLM-CN-DV). The role of vegetation feedback is examined
based on simulations with and without the DV module. Simulations from four different global climate models are used as
lateral boundary conditions (LBCs) for historical and future periods (i.e., historical: 1981–2000; future: 2081–2100). With
utilizing the Standardized Precipitation Evapotranspiration Index (SPEI), we quantify the frequency, duration and severity of
droughts over the focal regions of the Sahel, the Gulf of Guinea, and the Congo Basin. With the vegetation dynamics being
considered, future droughts become more prolonged and enhanced over the Sahel, whereas for the Guinea Gulf and Congo
Basin, the trend is opposite. Additionally, we show that simulated annual leaf greenness (i.e., the Leaf Area Index) well-
correlates with annual minimum SPEI, particularly over the Sahel, which is a transition zone, where the feedback between
land-atmosphere is relatively strong. Furthermore, we note that our findings based on the ensemble mean are varying, but
consistent among three different LBCs except for one LBC. Our results signify the importance of vegetation dynamics in
predicting future droughts in West Africa, where the biosphere and atmosphere interactions play a significant role in the
regional climate setup.
**1 Introduction**
West Africa is significantly vulnerable to climate change yet, projecting its future climate is a challenging task (Cook, 2008).
From the 1970s, a long period of drought was observed over West Africa, lasting until the late 1990s. While it is important to
reduce the uncertainties and improve the reliability of future climate projections, there is still no clear consensus about whether
the future outlook of the West African hydroclimate will be drier or wetter. Some studies projected drying trends (Hulme et
al., 2001), whereas others predicted a wetter future (Hoerling et al., 2006; Kamga et al., 2005; Maynard et al., 2002). Caminade
and Terray (2010) reviewed the A1B scenarios of the 21 coupled models from the Coupled Model Intercomparison Project



(CMIP) Phase 3 (CMIP3), which focused on a balanced emphasis on all energy resources, for the Sahel and found no clear
evidence of precipitation trending over Africa. Roehrig et al. (2013) combined the CMIP3 and CMIP Phase 5 (CMIP5) global
climate models (GCM) and found that western end of Sahel shows a drying trend whereas eastern Sahel shows opposite trend.
Limited-area models, i.e., Regional Climate Models (RCMs) are often used as they can capture finer details as compared to
GCMs (Kumar et al., 2008). The physics of RCMs dominate the signals imposed by large-scale forcing (i.e., forces with
boundary conditions derived from GCMs). However, discrepancies still remain, because RCMs have distinct systematic errors
with West African precipitation, varying in amplitude and pattern across models (Druyan et al., 2009; Paeth et al., 2011).
Because climate and greenhouse gas concentrations continuously change, a noticeable change in vegetation is
expected (Yu et al., 2014b). A more representative and reliable model requires incorporation of dynamic vegetation (DV)
instead of static vegetation (SV) (Alo and Wang, 2010; Patricola and Cook, 2010; Wramneby et al., 2010; Xue et al., 2012;
Zhang et al., 2014). Charney et al. (1975) first conceptualized the idea that precipitation could change dynamically in response
to vegetation variability, he claimed that changes in precipitation over the Sahel is due to reduction in vegetation and increase
in albedo. Various studies of biosphere–atmosphere interactions have been documented (Wang and Eltahir, 2000; Patricola
and Cook, 2008; Kim et al., 2007) but there are a few studies in which a coupled RCM-DV is used. Such studies are in their
initial stages (Cook and Vizy, 2008; Garnaud et al., 2015; Wang et al., 2016; Yu et al., 2016). For example, Cook and Vizy
(2008) introduced a coupled potential vegetation model into an RCM to estimate the influence of global warming on South
American climate and vegetation. They found a reduction in vegetation cover of almost 70% in the Amazon rainforest
highlighting the importance of using DV in RCMs. Recently, Wang et al. (2016) introduced a DV feature into the International
Center for Theoretical Physics Regional Climate Model (RegCM4.3.4) (Giorgi et al., 2012) with Carbon–Nitrogen (CN)
dynamics and DV (RegCM-CLM-CN-DV) of the community land model (CLM4.5) (Lawrence et al., 2011; Oleson et al.,
2010). They validated the coupled model over tropical Africa (Wang et al., 2016; Yu et al., 2016). The advantage of simulating
DV in the model eliminates potential discrepancies between the climate conditions and bioclimatic conditions required to
prescribed vegetation, but it can create climate draft, i.e., biases in the model (Erfanian et al., 2016). Additionally, such a model
is advantageous, because it provides a capacity to simulate future terrestrial ecosystems as the climate evolves.
Among various drought indices (e.g., the Palmer Draught Severity index (Palmer, 1965) and the Standard
Precipitation Index (McKee et al., 1993)) used to assess drought events, Vicente–Serrano (2010) suggested the Standardized
Precipitation Evapotranspiration Index (SPEI), which uses the deficit between precipitation and potential evapotranspiration.
Since the development of SPEI, various researchers have adopted this index for drought studies (Boroneant et al., 2011; Deng,
2011; Li et al., 2012a; Li et al., 2012b; Lorenzo–Lacruz et al., 2010; Paulo et al., 2012; Sohn et al., 2013; Spinoni et al., 2013;
Wang et al., 2016; Yu et al., 2014a; Yu et al., 2014b). Abiodun et al. (2013) studied the climate change and corresponding
extreme events caused by afforestation in Nigeria while defining the drought events using SPEI. McEvoy et al. (2012) used
SPEI as a drought index to monitor conditions over Nevada and Eastern California, proposing that SPEI was a convenient tool
to describe the drought in arid regions.





In this study, we aim to understand the impact of vegetation feedback on the future of droughts over West Africa.
Specifically, SPEI is used to depict vegetation feedback on drought characteristics according to frequencies, severity, and
duration over West Africa. Four sets of GCMs are used to force the RCM with and without vegetation dynamics. By comparing
the drought characteristics between the two simulation sets, we show the signals of DV on the drought processes in different
regions of Africa.

## 2 Methodology

### 2.1 Model Description

This study uses state-of-the-art RegCM-CLM-CN-DV (Wang et al., 2016). Specifically, RegCM4.3.4 (Giorgi et al., 2012) and
CLM4.5 (Lawrence et al., 2011; Oleson et al., 2010) with CN dynamics and DV are coupled to simulate various atmospheric,
land, biogeochemical, vegetation phenology, and vegetation distribution processes. RegCM is a regional model that uses an
Arakawa B-grid finite differencing algorithm along with a terrain-following σ-pressure vertical coordinate system. Grell et al.
(1994) introduced an additional dynamic component in the model that is taken from the hydrostatic version of the Pennsylvania
State University Mesoscale Model version 5. From Community Climate Model (Kiehl et al., 1996) a radiation scheme was
added. Model covers four different convection parameterization schemes namely 1) the modified-Kuo scheme (Anthes et al.,
1987), 2) the Tiedtke scheme (Tiedtke, 1989), 3) the Grell scheme (Grell, 1993) and 4) the Emanuel scheme (Emanuel, 1991)
along with non-local boundary layer scheme of Holtslag et al. (1990).Cloud and precipitation scheme comes from the physics
package (Pal et al., 2000). Aerosols algorithm follows Solmon et al. (2006) and Zakey et al. (2006).
While solving a surface biogeochemical, biogeophysical, ecosystem dynamical and hydrological processes, CLM4.5
considers fifteen soil layers, sixteen distinct plant functional types (PTF), up to five snow layers and a ordered data structure
in each grid cell (Erfanian et al., 2016; Lawrence et al., 2011; Wang et al., 2016). An optional component present in this model
is the CN and DV module. CN module not only simulates CN cycles and plant phenology and maturity but also estimates
vegetation height, stem area index and leaf area index (LAI). The DV module projects the fractional coverage of different
PFTs and corresponding temporary variations at yearly time steps developed using CN-estimated carbon budget, also it
accounts for plant existence, activity and formation. If CN and DV modules are inactive, it means that the distribution and
vegetation composition in the model is established according to observed data sets (i.e., SV).

### 2.2 Numerical Experiments

This study focuses on the West African region with emphasis on three regions over the study domain (see Fig. 1): the Sahel,
the Gulf of Guinea, and the Congo Basin. A total of 16 different numerical simulations are performed (Table 1). To investigate
the impacts of DV, simulation of model is carried out in two distinct configurations, one in which CN-DV module is activated
(i.e, DV runs) and the other in which CN-DV module is not activated (i.e., SV runs). Additionally, the LBCs for the RCMs
are derived from four GCMs: the Community Earth System Model (Kay et al., 2015), the Geophysical Fluid Dynamics





Laboratory model, the Model for Interdisciplinary Research on the Climate–Earth System Model (Watanabe et al., 2011), and
the Max Planck Institute Earth System Model. These eight simulations are performed for two different periods: the present
(i.e., 1981–2000) (CMIP5-historical) and the future (i.e., 2081–2100) (CMIP5-RCP8.5).
The model grid is configured using a 50-km horizontal grid spacing and 18 vertical layers from the surface to 50 hPa.
The model parameterizations are the same as the one used by Wang et al. (2016) and Yu et al. (2016), which was optimized
with previous applications over the same region (Alo and Wang, 2010; Saini et al., 2015; Wang and Alo, 2012; Yu et al.,
2014b). Its performance and simulation details with ERA-interim and future projections were documented by Wang et al.
(2016) and Erfanian et al. (2016), respectively.

**2.3 SPEI**

Vicente–Serrano et al. (2010) gave a simple approach to estimate SPEI. Thornthwaite (1948) method is used to calculate
monthly PET in first step, this method utilizes three parameters 1) temperature, 2) latitude and 3) time. For a given month, j,
and year, i, the monthly water surplus or deficit, $(D_{i,j})$ is calculated by Eq. (1) given below.
$$D_{i,j} = PR_{i,j} - PET_{i,j} \tag{1}$$
Where PR is precipitation and PET is potential evapotranspiration. In the second step accumulated monthly water
deficits, $(X_{i,j}^{k})$, at time scale $k$ (i.e., 12 months) in a given month, $j$, and year, $i$, is calculated based on $D$. Finally, $SPEI_{i,j}^{k}$ is
estimated by fitting $X_{i,j}^{k}$ to the log-logistic distribution by mean of the L-moments method by (Hosking 1990). In this study,
we define a drought event with an $SPEI_{i,j}^{k}$ of less than -1.

**3 Results and Discussions**

**3.1 Historical Climate, Vegetation and Drought**

This study briefly presents the present-day climate, vegetation, and droughts, simulated with RegCM-CN-DV with and without
vegetation dynamics, as detailed evaluations of model performance, including the performance according to different RCMs,
which was already provided by Erfanian et al. (2016). Relative to the observational data from the University of Delaware (Fig.
1), both SV and DV ensembles (Figs. 2a and 2b) follow the observed spatial patterns of precipitation and air temperature with
overestimating precipitation over Gulf of Guinea and the northern and southern parts of the Congo Basin. But over Sahel and
the central Congo Basin it is underestimated. The spatial trend of temperature bias is almost similar to precipitation bias, with
the dry and warm bias occur simultaneously and vice versa. It also reflects how evaporative cooling plays an important role in
surface energy flux across the regions (Erfanian et al., 2016). Additionally, the model generally performs better with SV than
with DV. The biases of precipitation and temperature in SV ensembles are further amplified in the DV ensembles. DV tends
to remove the physical inconsistencies linked with SV, but it increases the sensitivity of the model to lateral boundary
conditions (LBC) and potential model biases related to LBCs (Erfanian et al., 2016). So, we can say that one of the benefits to





introduce DV in the model is that it gives us a clear signal that how the change of vegetation could impact climate forcings,
presented in Sections 3.2 and 3.3.

27        By allowing vegetation dynamics, the LAI is overestimated in the Guinea Gulf and the central Congo Basin, and it is

underestimated in the Sahel region and southern and northern parts of the Congo Basin, compared to the case without
vegetation dynamics, where the LAI represents Moderate Resolution Imaging Spectroradiometer-based monthly-varying
climatological values (Figs. 3a, 3b, and 3e). It seems that underestimated LAI over the Sahel region is due to dry bias in the
atmospheric forcings, which then leads to additional decreases in precipitation for that region. Such dry biases lead to warm
bias in air temperate via the reduction of evaporative cooling.
The precipitation surplus/deficit (Eq. (1), Fig. 2c) was used in calculating SPEI values to analyze the drought frequency.
Precipitation minus potential evapotranspiration is mainly controlled by air temperature according to Thornthwaite method.
The difference of DV and SV ensembles for the precipitation surplus/deficit (Fig. 2c-3) follow that of the precipitation and
temperature, as expected.

37        Therefore, the difference for the drought frequency (Fig. 4a) depicts a similar pattern.  For historical period over Sahel

drought frequency is up to 44% higher when DV is enabled whereas it is 40% less over the Gulf of Guinea. Such characteristics
in the ensemble averages are captured in the difference of drought frequency between DV and SV of each ensemble member
to different extents (the first row of Fig. 5). While the Sahel and the Guinea Coast regions present relatively similar differences
in the drought frequency, the central Congo Basin shows quite different trends among the different LBCs.  CCSM presents
increase in drought frequency in DV relative to SV, but MIROC presents the opposite. GFDL and MPI-ESM presents relatively
weak differences.

44        To investigate the role of vegetation dynamics on drought severity and duration, the averages of SPEI over three

regions are estimated in Fig. 6. In the Sahel, the more severe and longer droughts are clearly captured for the present-day DV
ensemble compared to the SV ensemble. As noted, the reason behind an underestimated LAI over Sahel is dry biasness in
atmospheric forcings, which then leads to an additional decrease in precipitation in that region. Thus, prolonged and severe
drought events are consistently found in DV ensembles for Sahel. In the Guinea Coast and the Congo, the opposite is found
because of the vegetation dynamics. Also, different LBCs present consistent patterns except for CCSM, which shows limited
differences of SPEI between DV and SV in the regional averages over the Congo and Gulf of Guinea.
**3.2 Predicted Future Climate, Vegetation, and Droughts**
In this section, we focus on the projected future climate, vegetation, and droughts, simulated with and without vegetation
dynamics. First of all, projected precipitation in the future period of both SV and DV ensembles (Figs. 7a and 7b) shows the
similar spatial patterns to that of the past with different regional changes. In the SV ensemble (Fig. 7a-3), small decrease in
precipitation are found in Sahel and the Congo Basin. For the DV ensemble (Fig. 7a-4), it is clearly visible that the band of
precipitation below 10 °N increases up to 56.4 m/month. As expected, atmospheric warming caused by the increased $CO_2$





concentration in the future scenario leads to widespread increases in temperatures for both SV and DV ensembles (Figs. 7b-3
and 7b-4).
Consistent with such changes in climate conditions, vegetation state (i.e., LAI) changes because of atmospheric
warming and $CO_2$ fertilization. In the DV ensemble (Figs. 3d and 3e), widespread increases in future LAI are found, compared
to that from the historical period over the regions below 10 °N. Beyond 10 °N, vegetation cover is sparse and there are no
noticeable changes in future LAI. Note that LAI does not differ for both historical or future periods in SV.
In the future, the precipitation surplus/deficit shows a general decline for both SV and DV ensembles (Figs. 7c-3 and 7c-4).
Only local increases in precipitation surplus/deficit near 10 °N are captured by the DV ensemble. Such changes in precipitation
surplus/deficit lead to similar changes in drought frequencies between the future and historical periods for both SV and DV
ensembles (Figs. 4b and 4c). Corresponding to the band of precipitation increase, a slight decrease of drought frequency of up
to 15 % is shown in the DV ensemble.
**3.3 Impact of vegetation dynamics on future droughts**
It is desired to include vegetation dynamic component in land-atmospheric coupled model for future climate projections,
although including this property makes the model more complex but it is closer to a realistic model.  In this section, we focus
on the role of vegetation dynamics in future ensembles (i.e., the difference between DV and SV for the future).
Investigating the difference of LAI between DV and SV for the future period (Fig. 3f), we find that the LAI for the DV
ensemble is smaller than that of SV over the Sahel and larger below 10 °N. Such different responses of vegetation can be
attributed to dominant vegetation types over the regions as grasses and trees are dominant over the Sahel and below the 10°N
respectively. We note that LAI differences between SV and DV ensembles, show quite similar patterns both in historical and
future periods (Figs. 3c and 3f) with LAI biases caused by climate biases in the historical period being similarly shown in the
future period. Note that underestimated LAI in Sahel is not necessarily a bias in the future simulations, because the future LAI
in SV is assumed to be identical to historical climatological LAI as in historical SV ensemble.
Differences between DV and SV in precipitation and air temperature (Figs. 7a-5 and 7b-5) follow the differences of
the vegetation state (i.e., LAI). Over the region below 10 °N, wetter and colder climate conditions are predicted with the DV
ensemble compared to the SV ensemble, resulting in increased precipitation surplus, as shown in Fig. 7c-5. Consequently, the
frequencies of drought events decrease up to 40 % over Gulf of Guinea and increases up to 43 % over the Sahel based on the
ensemble averages (Fig. 4d). Among the runs with different LBCs, the inconsistency in the drought frequency is found over
the central Congo Basin with CCSM, as already pointed out in the historical simulations.
The differences of regional averages of SPEI over the three different regions (see the last rows in each panel of Fig.
6) present the impact of vegetation dynamics on future drought severity and duration. Ensemble averages show that more
prolonged and more severe droughts are projected over the Sahel and vice versa for the Guinea Gulf and the Congo Basin.
Among ensemble members with different LBCs, CCSM presents a bit different results from other LBCs, not capturing the
decreased droughts for the Guinea Gulf and the Congo Basin.





We next present the correlation coefficients between annual maximum LAI and annual minimum SPEI over the
regions for both historical and future periods (Fig. 8). With drought events, as reflected in the relatively lower annual minimum
SPEI, the annual maximum LAI should be smaller, because leaf growth is limited during such events. Such interactive
responses of vegetation to climate conditions are only captured in the DV ensemble. When DV is active, a large portion of
West Africa has a strong positive association between the maximum LAI and minimum SPEI. Relatively strong correlations
are found along the Sahel, which may attribute to the fact that feedback between land–atmosphere is relatively strong in
transition zones.

## 4 Conclusion

In this study, we employed the drought index (i.e., SPEI) to quantitatively assess the effects of vegetation dynamics on
projected future drought over West Africa. The impact of vegetation feedback on drought projection was examined both with
and without considering vegetation dynamics. This study suggests that, with the vegetation dynamics considered, drought is
prolonged and enhanced over the Sahel, whereas for the Guinea Gulf and Congo Basin, the trend is clearly the opposite. Such
opposite changes could attribute to amplified biases because a feedback exists between climate and vegetation in a dynamic
vegetation model, as well as due to bioclimatic inconsistency in the static vegetation model. These results are quite consistent
over 3 different LBCs while the LBC with CCSM show somewhat opposite results for the Congo Basin. Furthermore, we
show that simulated annual leaf greenness (i.e., LAI) was well correlated with annual minimum SPEI, particularly over the
Sahel, which is a sensitive, transition zone, where the feedback between land–atmosphere is relatively strong.
We note that the present study uses the SPEI via calculating PET with the Thornthwaite approach, that considers air
temperature as a governing feature of PET. There are various other method one of them is Penman method that that include
many other variables (i.e., humidity, radiation coefficient and wind speed) to calculate PET. Due to temperature rise, there
may be limited effects on drought via increased PET because other climatic conditions affecting PET may balance for
temperature rise (McVicar et al., 2012).
*Data Availability*. Observed data was collected from University of Delaware and model output data are available in
https://github.com/yjkim1028/RegCM-CN-DV_data. In addition, a map with the country boundaries is drawn with 'mapdata'
package of R-studio.
*Author contribution*. YK and GW designed the study and AE performed the simulations. MSM, JH and MU performed the
results analysis. MSM, YK, AE and GW wrote the manuscript.
*Competing interests*. The authors declare that they have no conflict of interest.
*Acknowledgements*. This study was supported by the Basic Science Research Program through the National Research
Foundation of Korea, which was funded by the Ministry of Science, ICT & Future Planning (2018R1A1A3A04079419) and
the Internationalization Infra Fund of Yonsei University (2018 Fall semester).



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

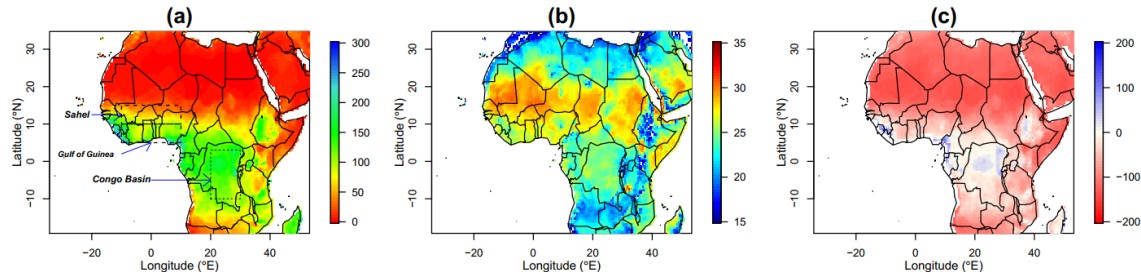

**Figure 1.** Observed averages of (a) precipitation (mm/month) and (b) air temperature ($^{o}$C) from 1981–2000 using datasets from the University of Delaware, and (c) derived precipitation deficit/surplus (mm/month). In (a), the boxes with the dashed lines show three focal regions of Sahel, Gulf of Guinea and the Congo Basin.

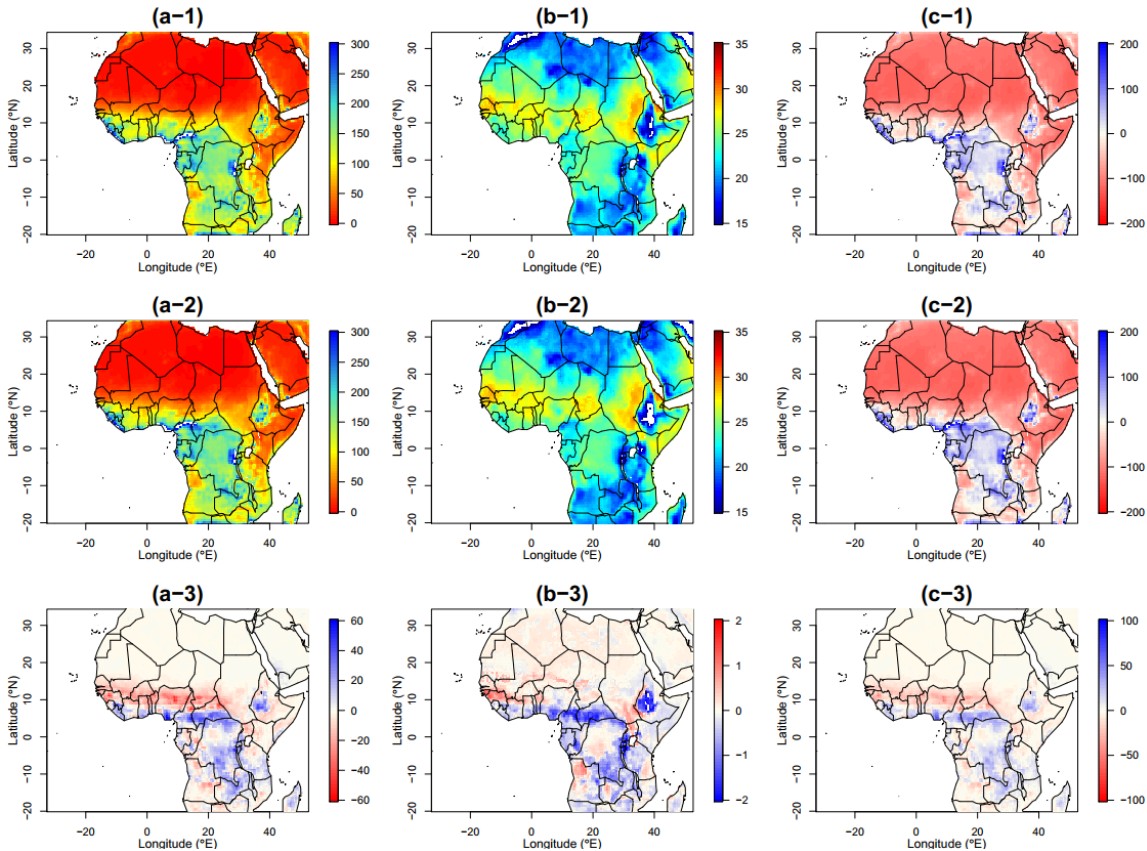

**Figure 2.** Averages of simulated (a) precipitation (mm/month), (b) temperature (°C), and (c) derived precipitation surplus/deficit (mm/month)

from 1) SV ensembles, 2) DV ensembles, and 3) the difference between DV and SV ensembles for the historical period of 1981–2000.



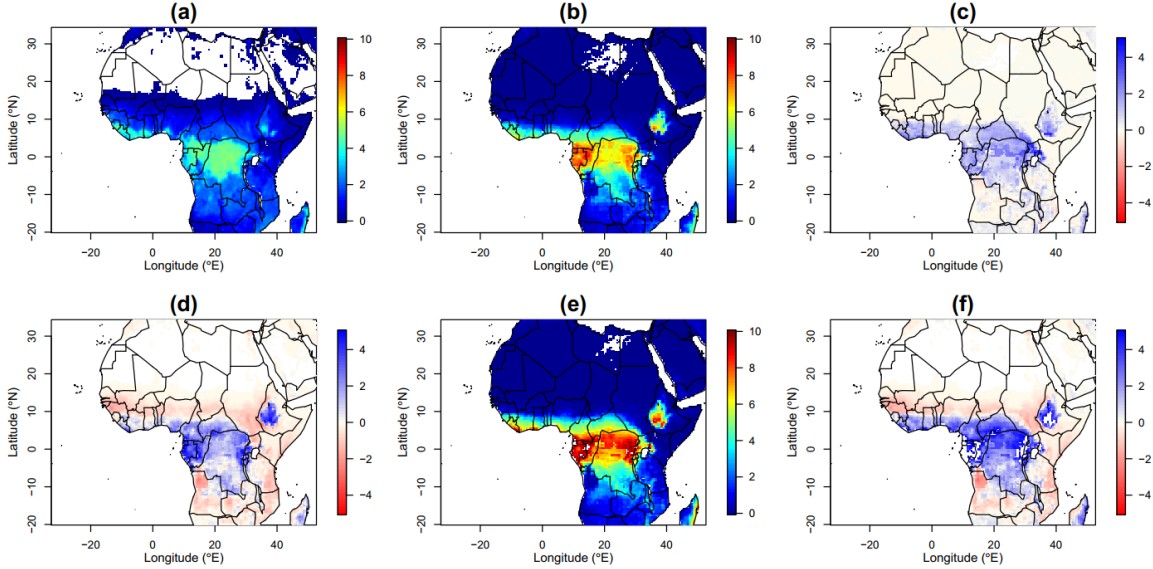

**Figure 3.** Averages of leaf area index (LAI) (a) used for SV and (b) simulated in DV ensembles for historical period (1981–2000) and (c) their differences (DV-SV). And, we show (d) the difference between future (2081-2100) and historical periods in DV, (e) averages of simulated LAI in DV ensembles for future period and (f) the difference between DV and SV in the future period.

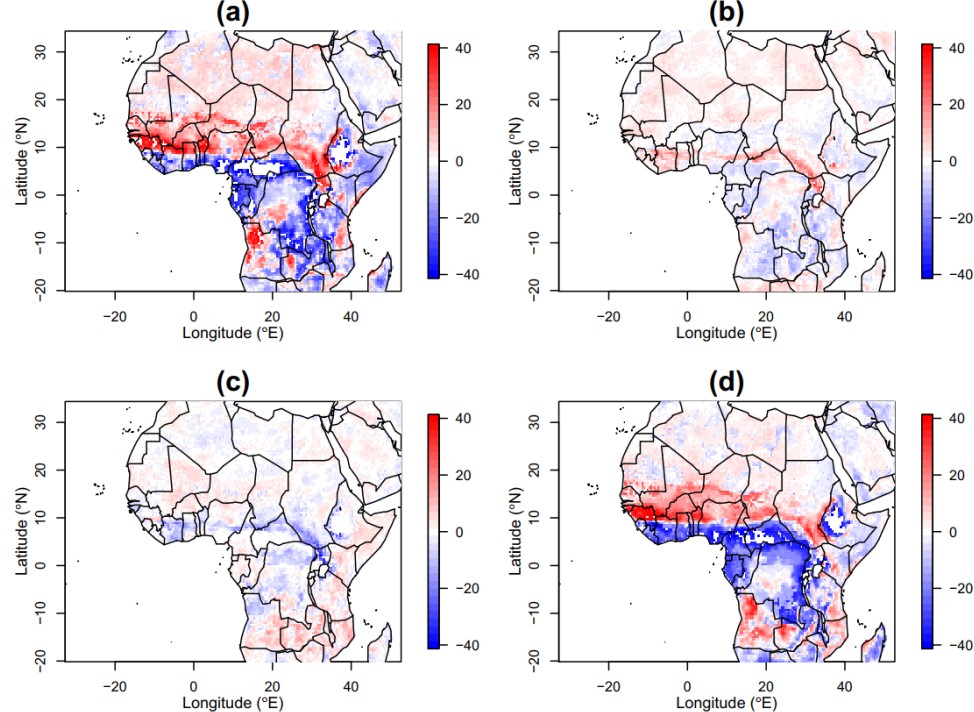

**Figure 4.** Difference of drought frequencies between the DV and the SV ensembles (a) for the historical period (1981-2000) and (d) for the future period (2081-2100). Differences between the future and historical periods (future-historical) for (b) SV ensembles and (c) DV ensembles. Drought frequency is defined for events with an SPEI less than -1.



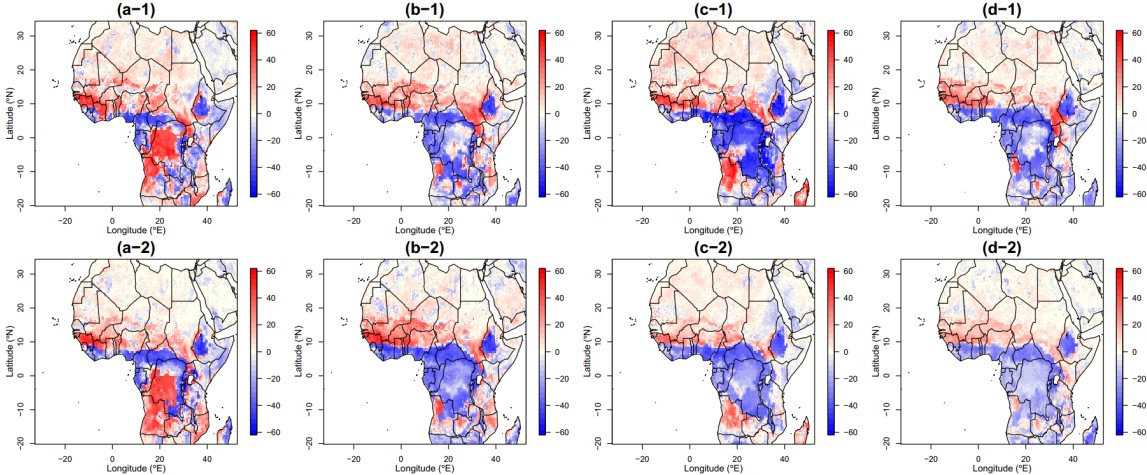

64

**Figure 5.** Difference of drought frequencies between the DV and the SV ensembles (1) for the historical period (1981-2000) and (2) for the

future period (2081-2100) from the ensemble members with different LBCs of (a) CCSM, (b) GFDL, (c) MIROC and (d) MPI-ESM. Drought

frequency is defined for events with an SPEI less than -1.









**Figure 6.** Monthly SPEI averaged for three regions of the Sahel, the Gulf of Guinea, and the Congo Basin in (a) ensembles and the individual
member with different LBCs of (b) CCSM, (b) GFDL, (c) MIROC and (d) MPI-ESM. HSV and HDV (FSV and FDV) represent the historical
(future) simulation without and with dynamic vegetation, respectively. HDV-HSV (FDV-FSV) depict the difference between HDV and HSV
(FDV and FSV).






**Figure 7.** Averages of simulated (a) precipitation (mm/month) and (b) temperature (°C), and (c) derived precipitation surplus/deficit
(mm/month) from 1) SV ensembles and 2) DV ensembles for the future period of 2081–2100. Their difference between future and historical
periods (future-historical) for 3) SV ensembles and 4) DV ensembles are shown. The difference between DV and SV ensembles reflect the
future period.





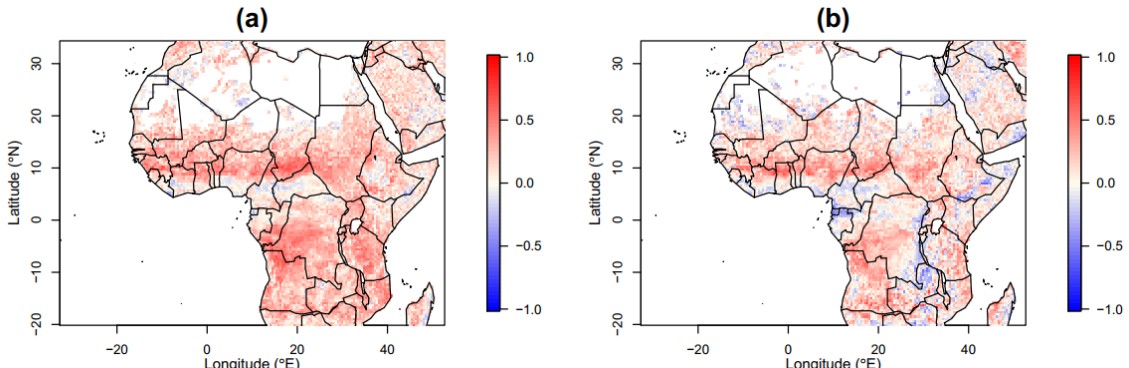

**Figure 8.** Spearman's rank correlation coefficient between annual minimum LAI and annual maximum SPEI from the DV ensembles for (a) the historical (1981-2000) and (b) future (2081-2100) periods.




**Table 1**. Description of 16 different simulation setups (4 boundary conditions, 2 different vegetation dynamics and 2 different periods)

| Boundary conditions from different GCMs | CCSM | Community Earth System Model |
|---|---|---|
| | GFDL | Geophysical Fluid Dynamics Laboratory |
| | MIROC | Model for Interdisciplinary Research on Climate-Earth System Model |
| | MPI-ESM | Max Planck Institute Earth System Model |
| Vegetation dynamics | DV | Dynamic Vegetation |
| | SV | Static Vegetation |
| Periods | Historical | 1981–2000 |
| | Future | 2081–2100 |


