# Peer review of "Projected effects of vegetation feedback on drought characteristics of 1 West Africa using a coupled regional land-vegetation-climate model 2"

_Hydrology and Earth System Sciences, 2019_

## Referee Comment (RC1) · Anonymous Referee #1 · 19 Aug 2019

There are several minor or major concerns for this paper. 1. Only observed precipitation, air temperature and precipitation deficit/surplus were shown in the Figure 1. How reliable are the simulations with and without the dynamic vegetation (DV)? Also, what's the performance of the simulated historical Leaf area index (LAI) by the DV version? 2. LAI has been used as one of the main indicators representing the responses of land surface to regional climate change. For the DV version, what will the plant functional types (PFTs) change with different climate scenarios? How may the regional climate changes be associated with the projected land surface changes including vegetation

(e.g., PFTs area extent and tree height) induced changes in energy and water fluxes? 3. Were any differences between the DV and SV or between the future and historical periods statistically different? 4. There is no any discussion in this paper. It's needed because people got to know why and how your present work is important and unique, and what the limitations and next steps are.

———————————————————

---

## Author Comment (AC1) · 19 Sep 2019

**Response to Reviewer #1**

There are several minor or major concerns for this paper.
1. Only observed precipitation, air temperature and precipitation deficit/surplus were shown in the Figure 1. How reliable are the simulations with and without the dynamic vegetation (DV)? Also, what's the performance of the simulated historical Leaf area index (LAI) by the DV version?

>> We understand it is critical to evaluate the model performance but this study builds upon previous studies (Wang et al., 2016; Yu et al., 2016; Erfanian et al., 2016) that documented model performances with and without the vegetation dynamics. As per the reviewer's comments, we clarified these points as follows:

Line 91: *"Wang et al. (2016) extensively evaluated the RegCM-CLM-CN-DV model for simulating regional climate and ecosystems in West Africa. It was performed using the lateral boundary conditions (LBCs) from the ERA-Interim, and with and without vegetation dynamics. Yu et al. (2016) and Erfanian et al. (2016) also examined the impacts of vegetation dynamics on the climate and ecosystems with multiple LBCs from past and future GCM simulations. Building upon these previous studies, this study investigates the impacts of vegetation dynamics on the regional drought characteristics, i.e., frequency, duration, and severity over the focal regions of the West African domain: the Sahel, the Gulf of Guinea, and the Congo Basin (Fig. 1)."*

Line 118: *"This study briefly presents the present-day climate, vegetation, and droughts simulated with RegCM-CLM-CN-DV with and without vegetation dynamics, based on detailed evaluations of model performance in previous studies with the same model, i.e., Wang et al. (2016), Yu et al. (2016) and Erfanian et al. (2016)."*

Regarding the performance of LAI, Fig. 3c presents the LAI difference between DV and SV. We have revised the paragraph to clarify this as follows:

Line 152: *"With the addition of vegetation dynamics, the LAI is overestimated in the Guinea Gulf and the central Congo Basin, and it is underestimated in the Sahel region and southern and northern parts of the Congo Basin, whereas in the case without vegetation dynamics, the LAI represents Moderate Resolution Imaging Spectroradiometer-based monthly-varying climatological values (Figs. 3a, 3b, and 3e). Over the Sahel, the model underestimates the woody plants and grasses with significant overestimation of the bare ground area, which can be attributed to biases in the CLM-CN-DV as well as the RCM physical climate, i.e., dry bias (Wang et al., 2016; Erfanian et al., 2016)."*

2. LAI has been used as one of the main indicators representing the responses of land surface to regional climate change. For the DV version, what will the plant functional types (PFTs) change with different climate scenarios? How may the regional climate changes be associated with the projected land surface changes including vegetation (e.g., PFTs area extent and tree height) induced changes in energy and water fluxes?

>> As per the reviewer's suggestion, we have revised the manuscript to relate changes of LAI with changes of vegetation types as well as the surface conditions that follow. However, note that those have been well documented in our previous studies (Wang et al., 2016; Yu et al., 2016; Erfanian et al., 2016; refer to Fig. R1), so we have added the explanations without additional figures to keep our focus on drought characteristics.

Line 210: *"As examined in Erfanian et al. (2016), along with lower LAI in the Sahel, with DV in comparison to SV, higher albedo, lower cooling, lower evapotranspiration, and lower precipitation is simulated as strong land-atmosphere coupling is known in the region like Sahel. Also note that such changes in LAI do not always accompany changes in the dominant vegetation types. In the Sahel, there will be more grass in future with increased LAI, and changes in land cover from grasses to woody plants are found in the Gulf of Guinea."*

[Figure]

**Figure 7.** Future changes of annual fractional coverage (%) of (a) woody plants, (b) grasses, (c) bare ground, and (d) future changes of annual LAI for MME of RCM-CLM-CNDV simulations as of 2081–2100 compared with 1981–2000. Shading is applied only to areas where changes pass the 1% significance test.

*Fig. R1. Changes of vegetation fractions in the future (Erfanian et al., 2016)*

3. Were any differences between the DV and SV or between the future and historical periods statistically different?

>> As per reviewer's comments, we have added dots to the difference figures (Fig. 2, 3, 4, 5 and 7) to identify the statistically significant differences, with an example of Fig. 3 below.

[Figure]

*Figure 3. Averages of leaf area index (LAI) (a) used for SV and (b) simulated in DV ensembles for the historical period (1981–2000), and (c) their differences (DV-SV). And, we show (d) the difference between future (2081-2100) and historical periods in DV, (e) averages of simulated LAI in DV ensembles for future period, and (f) the difference between DV and SV in the future period. Doted regions show areas passing the two-tailed confidence level with α=0.01.*

4. There is no any discussion in this paper. It's needed because people got to know why and how your present work is important and unique, and what the limitations and next steps are.

>> We have expanded the Conclusions and Discussion sections with discussion about the importance and limits of this study as well as future study directions.

Line 251: *"While most future drought characterization studies with climate model predictions have been carried out without considering the role of vegetation (e.g., Cook et al., 2015; Huang et al, 2018), this study suggests the necessity of the comprehensive understanding of biosphere–atmosphere interactions in future drought projections. Furthermore, it has been pointed out that such land–atmosphere feedbacks could exacerbate droughts under future climate projections (Dirmeyer et al., 2013; Zhou et al., 2019). Therefore, these drought studies could be critical over not only the Sahel but also over other regions where positive feedbacks between land and atmosphere are strong, such as the interior of North America (Kim and Wang, 2007; Wang et al., 2007).*

*The present study uses the SPEI via calculating PET with the Thornthwaite approach, which considers air temperature as a governing feature of PET. However, there are various other methods to calculate PET, and among them, the Penman–Montieth method could be another candidate that could be employed for the SPEI because it includes many other variables (i.e., humidity, radiation coefficient, and wind speed) to calculate PET. Other climatic conditions affecting PET may balance temperature rise (McVicar et al., 2012), and thus, further investigations with multiple*

*approaches could shed a light on future drought characteristics.*

*This study points out the potentially prolonged and enhanced drought events over the Sahel. In addition, many African countries are expected to experience population growth, and a majority of the population increase rate is found in Niger and Chad, which are neighboring countries in the Sahel (Ahmadalipur et al., 2019). Combined with the high likelihood of prolonged and enhanced drought, population growth and, thus, the associated increase in water demand will likely exacerbate the risks of future drought and will present challenges for climate change adaptation for managing water needs in the region."*

[revised manuscript text omitted]

Wang et al. (2016) extensively evaluated the RegCM-CLM-CN-DV model for simulating regional climate and ecosystems in West Africa. It was performed using the lateral boundary conditions (LBCs) from the ERA-Interim, and with and without vegetation dynamics. Yu et al. (2016) and Erfanian et al. (2016) also examined the impacts of vegetation dynamics on the climate and ecosystems with multiple LBCs from past and future GCM simulations. Building upon these previous studies, this

Moved (insertion) [1]

Moved (insertion) [2]

study investigates the impacts of vegetation dynamics on the regional drought characteristics, i.e., frequency, duration, and severity over the focal regions of the West African domain, the Sahel, the Gulf of Guinea, and the Congo Basin (Fig. 1).

As presented in Erfanian et al. (2016), a total of 16 different numerical simulations are performed (Table 1).

Numerical simulation is carried out in two distinct configurations, one in which CN-DV module is activated (i.e, DV runs)

and the other in which CN-DV module is not activated (i.e., SV runs). Additionally, the LBCs for the RCMs are derived from four GCMs: the Community Earth System Model (Kay et al., 2015), the Geophysical Fluid Dynamics Laboratory model, the

Model for Interdisciplinary Research on the Climate–Earth System Model (Watanabe et al., 2011), and the Max Planck

Institute Earth System Model. These eight simulations are performed for two different periods: the present (i.e., 1981–2000)

(CMIP5-historical) and the future (i.e., 2081–2100) (CMIP5-RCP8.5). The model grid is configured using a 50-km horizontal grid spacing and 18 vertical layers from the surface to 50 hPa. The model parameterizations are the same as the one used by previous studies of Wang et al. (2016), Yu et al. (2016) and Erafnian et al. (2016).

**2.3 SPEI**

Vicente–Serrano et al. (2010) gave a simple approach to estimate SPEI. Thornthwaite (1948) method is used to calculate monthly PET in first step, this method utilizes three parameters 1) temperature, 2) latitude and 3) time. For a given month, j, and year, i, the monthly water surplus or deficit, $(D_{i,j})$ is calculated by Eq. (1) given below.

$$D_{i,j} = PR_{i,j} - PET_{i,j} \tag{1}$$

Where PR is precipitation and PET is potential evapotranspiration. In the second step accumulated monthly water deficits, $(X_{i,j}^k)$, at time scale $k$ (i.e., 12 months) in a given month, $j$, and year, $i$, is calculated based on $D$. Finally, $SPEI_{i,j}^k$ is estimated by fitting $X_{i,j}^k$ to the log-logistic distribution by mean of the L-moments method by (Hosking 1990). In this study, we define a drought event with an $SPEI_{i,j}^k$ of less than -1.

**3 Results Analysis**

**3.1 Historical Climate, Vegetation and Drought**

This study briefly presents the present-day climate, vegetation, and droughts simulated with RegCM-CLM-CN-DV with and without vegetation dynamics, based on detailed evaluations of model performance in previous studies with the same model, i.e., Wang et al. (2016), Yu et al. (2016) and Erfanian et al. (2016). Relative to the observational data from the University of

Delaware (Fig. 1), both SV and DV ensembles (Figs. 2a and 2b) follow the observed spatial patterns of precipitation and air temperature with overestimating precipitation over Gulf of Guinea and the northern and southern parts of the Congo Basin.

But over Sahel and the central Congo Basin it is underestimated. The spatial trend of temperature bias is almost similar to precipitation bias, with the dry and warm bias occur simultaneously and vice versa. It also reflects how evaporative cooling plays an important role in surface energy flux across the regions (Erfanian et al., 2016). Additionally, the model generally performs better with SV than with DV. The biases of precipitation and temperature in SV ensembles are further amplified in
* * *
**Moved up [1]:** Yu et al.

**Moved down [3]:** Wang et al.

**Moved up [2]:** (2016) and Erfanian et al.

**Moved (insertion) [3]**

the DV ensembles. DV tends to remove the physical inconsistencies linked with SV, but it increases the sensitivity of the model to lateral boundary conditions (LBC) and potential model biases related to LBCs (Erfanian et al., 2016). So, we can say that one of the benefits to introduce DV in the model is that it gives us a clear signal that how the change of vegetation could impact climate forcings, presented in Sections 3.2 and 3.3.

With the addition of vegetation dynamics, the LAI is overestimated in the Guinea Gulf and the central Congo Basin, and it is underestimated in the Sahel region and southern and northern parts of the Congo Basin, whereas in the case without vegetation dynamics, where the LAI represents Moderate Resolution Imaging Spectroradiometer-based monthly-varying climatological values (Figs. 3a, 3b, and 3c). Over the Sahel, the model underestimates the woody plants and grasses with significant overestimation of the bare ground area, which can be attributed to biases in the CLM-CN-DV as well as the RegCM

physical climate, i.e., dry bias (Wang et al., 2016; Erfanian et al., 2016). Such dry biases lead to warm bias in air temperate via the reduction of evaporative cooling.

[revised manuscript text omitted]

While most future drought characterization studies with climate model predictions have been carried out without considering the role of vegetation (e.g., Cook et al., 2015; Huang et al, 2018), this study suggests the necessity of the comprehensive understanding of biosphere–atmosphere interactions in future drought projections. Furthermore, it has been pointed out that such land–atmosphere feedbacks could exacerbate droughts under future climate projections (Dirmeyer et al., 2013; Zhou et al., 2019). Therefore, these drought studies could be critical over not only the Sahel but also over other regions where positive feedbacks between land and atmosphere are strong such as the interior of North America (Kim and Wang, 2007).

The present study uses the SPEI via calculating PET with the Thornthwaite approach, which considers air temperature as a governing feature of PET. However, there are various other methods to calculate PET, and among them, the Penman-Montieth method could be another candidate that could be employed for the SPEI because it includes many other variables (i.e., humidity, radiation coefficient, and wind speed) to calculate PET. Other climatic conditions affecting PET may balance temperature rise (McVicar et al., 2012), and thus, further investigations with multiple approaches could shed a light on future drought characteristics.

This study points out the potentially prolonged and enhanced drought events over the Sahel. In addition, many African countries are expected to experience population growth, and a majority of the population increase rate is found in Niger and Chad, which are neighbouring countries in the Sahel (Ahmadalipour et al., 2019). Combined with the high likelihood of prolonged and enhanced drought, population growth and, thus, the associated increase in water demand will likely exacerbate the risks of future drought and will present challenges for climate change adaptation for managing water needs in the region. As examined in Erfanian et al. (2016), along with lower LAI in the Sahel, with DV in comparison to SV, higher albedo, lower cooling, lower evapotranspiration, and lower precipitation is simulated as strong land-atmosphere coupling is known in the region like Sahel. Also note that such changes in LAI do not always accompany changes in the dominant vegetation types. In the Sahel, there will be more grass in future with increased LAI, and changes in land cover from grasses to woody plants are found in the Gulf of Guinea.

*Data Availability*. Observed data was collected from University of Delaware and model output data are available in https://github.com/yjkim1028/RegCM-CN-DV_data. In addition, a map with the country boundaries is drawn with 'mapdata' package of R-studio.

*Author contribution*. YK and GW designed the study and AE performed the simulations. MSM, JH and MU performed the results analysis. MSM, YK, AE and GW wrote the manuscript.

*Competing interests*. The authors declare that they have no conflict of interest.

*Acknowledgements.* This study was supported by the Basic Science Research Program through the National Research Foundation of Korea, which was funded by the Ministry of Science, ICT & Future Planning (2018R1A1A3A04079419) and the Internationalization Infra Fund of Yonsei University (2018 Fall semester).

[revised manuscript text omitted]

---

## Referee Comment (RC2) · Anonymous Referee #2 · 22 Sep 2019

**1. Brief summary of the manuscript**

In their manuscript, Dr. Mehboob and co-workers applied a regional climate model coupled to a dynamic vegetation module to quantify the effects of vegetation feedback on drought over West (Sahel and Gulf of Guinea) and Central Africa (Congo Basin) under present-day and future climate. To identify drought conditions, the authors use the Standardized Precipitation Evapotranspiration Index (SPEI) as defined by Vicente-Serrano et al. (2010) by combining monthly precipitation and potential evapotranspiration (PET). To assess the added value of representing the dynamics of vegetation

processes (e.g., plant shift, growth), Mehboob et al. performed numerical experiments with and without the dynamic vegetation module. In addition, they accounted for uncertainties in the atmospheric forcing by taking boundary lateral conditions from four global climate models (GCMs). The main results are:

- In experiments using the dynamic vegetation module, future drought lengthens and strengthens in the Sahel compared to experiments without the dynamic vegetation module, while the trend is less clear in the Gulf of Guinea and the Congo Basin.

- When forcing the regional climate model with different GCMs, results are consistent except for the Congo Basin where GCM diverge in reproducing drought frequency under present-day and future climate.

**2. General comments**

The study addresses relevant scientific questions that are within the scope of HESS and that are related to drought occurrence and intensity in a sensitive region such as West and Central Africa. In this sense, the study could provide interesting advance towards current knowledge and methodologies applied to project drought in Africa and other sensitive regions using RCMs. However, in my opinion, the quality of presentation is poor and confused; the Introduction, Methodology, and Results and Discussion Sections are not well laid out; some methodological choices are not well justified; and the significance of results is not discussed. Moreover, I would suggest to edit and proofread the manuscript to avoid redundancy and to simplify some confused sentences that make the reading difficult. In the following, I provide specific comments (major and minor) on the manuscript.

**3. Major comments**

In my opinion, the **Introduction** does not provide enough information to readers on the

target region, its climate features (also in terms of surface-atmosphere interactions) and on the vegetation feedback the manuscript will focus on. Although the authors cite some previous works that studied the same region, I think the authors should spend more words in summarizing the main results and limits of the cited works. This will allow the authors to clearly state their own original contribution to the tackled topic.

**Ll 31 (pag. 2)**: "... on a balanced emphasis on all energy resources...": It is not clear to me what this mean. I suggest to rephrase this sentence and describe more explicitly the methodology of the cited work of Caminade and Terray (2010).
**Ll 36 (pag. 2)**: For sake of completeness, I would mention that RCM can be forced using re-analysis
**Ll 45–48 (pag. 2)**: I think it would be interesting to summarize the main findings of the study of Cook and Vizy (2008), in particular the effects on the regional climate of South America of a reduction of 70
bf Ll 53 (pag. 2): "...climate draft...": Again, this expression is unclear to me, I suggest to express this differently.
**Ll 55–63 (pag. 2)**: In my opinion, it is not clear why the authors have chosen the SPEI instead of other drought indexes. I would suggest to present the advantages and the limits of using the SPEI to identify and project drought.

In the **Methodology** section, I think the description of the dynamic vegetation module and its functioning should be more detailed. Moreover, I do not understand which parameterization scheme the authors have chosen to represent convection. Related to this point, to ensure the traceability of results, a summary table with all the selected parameterizations could be useful for readers that would like to apply the same modelling set-up over a different region.
In terms of run experiments, in my opinion, the study lacks an experiment forced by re-analysis; this extra-experiment would provide a better term of comparison against observations to identify the model biases.
Regarding the SPEI index, I think its computation should be described in a clearer

way. For example, the Thornthwaite method should be presented in more details to allow the readers to understand how the potential evapotranspiration is derived. Specifically, this method should also be shortly reviewed in comparison to other well-known methods (e.g., the Penman- Monteith equation), in a more detailed way than that reported on page 7 (ll. 7–11). Lastly, in the manuscript, the authors refer to drought frequency. However, it seems to me that they did not explicitly define how drought frequency has been calculated.

**Ll 82 (pag. 3)**: "... aN ordered data structure ...", it is not clear to me what this refers to. I would suggest to make this explanation more explicit.

In my opinion, in the **Results and Discussions** section, the model evaluation should be performed using a simulation forced by re-analyses. In the model evaluation presented in the manuscript, it is difficult to understand how the divergent behavior of GCMs over the Congo Basin may influence the ensemble mean, which is compared to observations in Figure 2. In general, I found the presentation and discussion of results confused and hard to follow using the provided figures. My suggestion would be to (a) re-structure this section and the related figures, (b) include a more quantitative discussion in relation to other studies, and (c) asses the significance of the shown results.

**Ll. 15 (pag. 4)**: "... different RCMs ...", by checking the study of Erfanian et al. (2016), I think the authors are referring to different GCMs.
**Ll. 18 (pag. 4)**: "... overestimating precipitation ...", it is hard to compare the figures and to distinguish the differences between observations and simulations, however it seems to me that precipitations are under-estimated over the Gulf of Guinea and the Congo Basin. A plot showing the differences between observations and model experiments will ease the identification and interpretation of model bias.
**Ll. 25–26 (pag. 5)**: This sentence is not clear to me. In RCM experiments, the climate forcing is prescribed, hence I do not understand how "a change in vegetation could

impact climate forcings".

**Ll. 45–46 (pag. 5)**: It is not clear to me that the experiments using the dynamic vegetation module clearly capture the "more severe and longer droughts". I think to support this statement an observation-based SPEI would be needed. If the authors could compute SPEI based on observations, I would suggest to add a line in Figure 6 that shows the monthly observation-based SPEI.

**Ll. 35 (pag. 5)**: " (Fig. 2c-3)" It is not clear to me if the authors are referring to Figure 2c and the whole Figure 3 or to something else.

In my opinion, the **figures** are not well laid out because title and units are only inserted in the figure caption. Since all the figures are multi-panel, the reading becomes even more complex. Moreover, in Figure 1 the three boxes are nearly invisible. I would suggest to highlight better the three target regions and to draw these boxes on all the maps that are presented in the study.

**4. Minor comments**

Below, I list typos and errors, and I point to sentences that I would suggest to rephrase in a clearer way.

**LL 14–15 (pag. 1)**: I would suggest to replace "With utilizing ..." with "Using ..."
**LL 16–17 (pag. 1)**: I would suggest to replace "With the vegetation dynamics ..." with "By considering vegetation dynamics ..."
**LL 33 (pag. 2)**: "... that western end of Sahel ... whereas eastern Sahel..." should be replaced with "...that the western end of Sahel ... whereas the eastern Sahel ..."
**LL 36 (pag. 2)**: I would suggest to remove the comma between "... remain ..." and "... because ..."
**LL 42 (pag. 2)**: "... variability, he claimed ..." should be replaced with "... variability; the authors claimed ..."
**LL 43 (pag. 2)**: "Various studies ... have been documented ..." should be replaced with

"... Various studies documented biosphere-atmosphere interactions ..."

**LL 51–54 (pag. 2)**: I would suggest to rephrase these two sentences to make them clearer and avoid redundancy.

**LL 55 (pag. 2)**: "...Draught ..." should be replaced with "... Drought ..."

**LL 57 (pag. 2)**: "..., which ..." should be replaced with "... that ..."

**LL 79 (pag. 3)**: A space is missing before "Cloud"

**LL 81 (pag. 3)**: I would suggest to correct and simplify this expression: " While solving a surface biogeochemical, biogeophysical, ecosystem dynamical and hydrological processes ..."

**LL 88 (pag. 3)**: "... distribution and vegetation distribution ... is established ..." should be replaced with "... distribution and vegetation distribution ... are established ... "

**LL 91–93 (pag. 3)**: I would suggest to rephrase the sentences that describe the different simulations to make them clearer and avoid redundancy.

**Ll 05 (pag. 4)**: The acronym PET has not been previously introduced.

**Ll. 56 (pag. 5)**: "CO2" should be replaced with "$CO_2$

**Ll. 75 (pag. 6)**: The comma between "ensembles" and "show" should be removed because it divides the subject from the verb.

**Ll. 35 (pag. 7)**: "... CCSM show somewhat ..." should be replaced with "... CCSM shows somewhat ..."

**Ll. 08 (pag. 7)**: There is an extra "that" which needs to be removed

---

## Author Comment (AC2) · 22 Nov 2019

**Response to Reviewer #2**

**1. Brief summary of the manuscript**

In their manuscript, Dr. Mehboob and co-workers applied a regional climate model coupled to a dynamic vegetation module to quantify the effects of vegetation feedback on drought over West (Sahel and Gulf of Guinea) and Central Africa (Congo Basin) under present-day and future climate. To identify drought conditions, the authors use the Standardized Precipitation Evapotranspiration Index (SPEI) as defined by Vicente-Serrano et al. (2010) by combining monthly precipitation and potential evapotranspiration (PET). To assess the added value of representing the dynamics of vegetation processes (e.g., plant shift, growth), Mehboob et al. performed numerical experiments with and without the dynamic vegetation module. In addition, they accounted for uncertainties in the atmospheric forcing by taking boundary lateral conditions from four global climate models (GCMs). The main results are:

- In experiments using the dynamic vegetation module, future drought lengthens and strengthens in the Sahel compared to experiments without the dynamic vegetation module, while the trend is less clear in the Gulf of Guinea and the Congo Basin.
- When forcing the regional climate model with different GCMs, results are consistent except for the Congo Basin where GCM diverge in reproducing drought frequency under present-day and future climate.

**2. General comments**

The study addresses relevant scientific questions that are within the scope of HESS and that are related to drought occurrence and intensity in a sensitive region such as West and Central Africa. In this sense, the study could provide interesting advance towards current knowledge and methodologies applied to project drought in Africa and other sensitive regions using RCMs. However, in my opinion, the quality of presentation is poor and confused; the Introduction, Methodology, and Results and Discussion Sections are not well laid out; some methodological choices are not well justified; and the significance of results is not discussed. Moreover, I would suggest to edit and proofread the manuscript to avoid redundancy and to simplify some confused sentences that make the reading difficult. In the following, I provide specific comments (major and minor) on the manuscript.

>> Thank you very much for the constructive comments. Here is the summary of our revision:

- 1) Add the model evaluations with the runs forced by the reanalysis data (ERA-Interim);
- 2) Re-arrange the results to better present our findings with updated figures;
- 3) Include additional and detailed literature review;
- 4) Discuss the significance on the results;
- 5) Improve the figures with re-arrangement and proper titles; Add the significant test results in the difference figures.

We have also proofread the manuscript thoroughly to avoid any confusing expressions. Please find the point-by-point responses to the specific comments below along with the revised manuscript.

**3. Major comments**

In my opinion, the **Introduction** does not provide enough information to readers on the target region, its climate features (also in terms of surface-atmosphere interactions) and on the vegetation feedback the manuscript will focus on. Although the authors cite some previous works that studied the same region, I think the authors should spend more words in summarizing the main results and limits of the cited works. This will allow the authors to clearly state their own original contribution to the tackled topic.

>> We have revised the introduction to include the additional and detailed literature review regarding the previous studies about the West African climate projects as well as the coupled climate-vegetation model development.

Page 2, Line 6: "Recently, Akinsanola and Zhou (2019) investigated projected changes in extreme summer rainfall events over West Africa with data from the Coordinated Regional Climate Downscaling Experiment (CORDEX) models. Results showed the RCMs reasonably reproduced the observed pattern of extreme rainfall over the region. Future projections under the representative concentration pathways (RCPs) showed a statistically significant decrease in total rainfall and an increase in consecutive dry days and extreme rainfall."

Page 2, Line 22: "Cook and Vizy (2008) developed a vegetation model coupled with a RCM to estimate the influence of global warming on South America by allowing interactions between climate and vegetation. With the simulation of the future climate under the A2 scenario, the authors found a reduction in vegetation cover of almost 70% in the Amazon rainforest along with a widespread increase in grass and shrubland in the east by the end of  $21^{st}$  century. This highlights the importance of considering vegetation dynamics in RCMs. Garnaud et al. (2015) combined the Canadian Regional Climate Model (CRCM5) with the Canadian Territorial *Ecosystem Model (CTEM) to investigate the impact of a vegetation model to simulate the present* day climate over North America. The result showed that introducing vegetation dynamics improved the model's performance in some regions, along with introducing new biases in other regions, owing to biases in simulated leaf area index (LAI). This atmospheric-vegetation interaction also introduced long term memory, which was estimated using a lagged correlation between temperature/precipitation and LAI. Wu et al. (2016) utilized a regional earth system model coupled with the dynamic vegetation model, RCA-GUESS (Smith et al., 2011), and investigated the role of vegetation dynamics on climate in Africa under the RCP8.5 projected climate scenario. The authors showed that introducing vegetation processes amplifies the warming trend and enhanced precipitation reduction over rainforest areas, which highlights the impact of introducing vegetation processes in a climate model."

Page 3, Line 3: "Recently, Wang et al. (2016) introduced a dynamic vegetation feature into the International Center for Theoretical Physics Regional Climate Model (RegCM4.3.4) (Giorgi et al., 2012) with carbon–nitrogen (CN) dynamics and dynamic vegetation (DV) (RegCM-CLM-CN-DV) of the community land model (CLM4.5) (Lawrence et al., 2011; Oleson et al., 2010) and validated the coupled model over tropical Africa. With the RegCM-CLM-CN-DV, Yu et al. (2016) and Erfanian et al. (2016) examined the impacts of vegetation dynamics on the climate and ecosystems using multiple LBCs from past and future GCM simulations over West Africa. Yu et al. (2016) showed that climate projections of dynamic vegetation feedback was found mainly in semiarid areas of West Africa with little signal in the wet tropics. Erfanian et al. (2016) demonstrated the substantial sensitivity of the simulated precipitation, evapotranspiration, and

soil moisture to vegetation representation. Including DV in the model eliminates potential inconsistencies between prescribed vegetation and climate, but it can cause climate drift (enhancing model biases) (Erfanian et al., 2016)."

Ll 31 (pag. 2): "... on a balanced emphasis on all energy resources...": It is not clear to me what this mean. I suggest to rephrase this sentence and describe more explicitly the methodology of the cited work of Caminade and Terray (2010).

>> As per the reviewer's suggestion, the review on Caminade and Terray (2010) has been rephrased to clarify their methodology and results.

Page 1, Line 28: "Caminade and Terray (2010) examined the simulated rainfall over the Sahel at the end of twenty-first century with the 21 models from the Coupled Model Intercomparison Project (CMIP) Phase 3 (CMIP3). They argued that different model projections are highly uncertain because future rainfall may be affected by changes in surface conditions (e.g., vegetation, land use and soil moisture) that have not been considered in CMIP3 models."

Ll 36 (pag. 2): For sake of completeness, I would mention that RCM can be forced using reanalysis.

>> As per the reviewer's suggestion, we have added the phrase about RCM.

Page 2, Line 6: "regional climate models (RCMs), which are forced with lateral boundary conditions (LBCs) derived from GCMs,"

Ll 45–48 (pag. 2): I think it would be interesting to summarize the main findings of the study of Cook and Vizy (2008), in particular the effects on the regional climate of South America of a reduction of 70.

>> As per the reviewer's suggestion, we have rephrased the sentences to clarify the findings of Cook and Vizy (2008).

Page 2, Line 22: "Cook and Vizy (2008) developed a vegetation model coupled with a RCM to estimate the influence of global warming on South America by allowing interactions between climate and vegetation. With the simulation of the future climate under the A2 scenario, the authors found a reduction in vegetation cover of almost 70% in the Amazon rainforest along with a widespread increase in grass and shrubland in the east by the end of 21st century. This highlights the importance of considering vegetation dynamics in RCMs."

Ll 53 (pag. 2): "...climate draft...": Again, this expression is unclear to me, I suggest to express this differently.

>> We have revised it to *"climate drift"* in the revised manuscript.

Ll 55–63 (pag. 2): In my opinion, it is not clear why the authors have chosen the SPEI instead of other drought indexes. I would suggest to present the advantages and the limits of using the SPEI

to identify and project drought.

>> As per the reviewer's suggestion, we have clarified the advantage of SPEI in Introduction. Further, recent studies on the estimation of the potential evapotranspiration have been discussed in Discussion and Conclusions.

Page 3, Line 13: "Various drought indices (e.g., the Palmer Drought Severity index (Palmer, 1965) and the Standard Precipitation Index (SPI, McKee et al., 1993)) have been used to assess Vicente–Serrano (2010) suggested the standardized precipitation drought events. evapotranspiration index (SPEI). It uses the deficit between precipitation and potential evapotranspiration and can include the effects of temperature variability on drought assessment. Therefore, it can be closely related to hydrologic and ecological drought processes although it only uses climate conditions. Since the development of SPEI, various drought studies have adopted this index (Boroneant et al., 2011; Deng, 2011; Li et al., 2012a; Li et al., 2012b; Lorenzo-Lacruz et al., 2010; Paulo et al., 2012; Sohn et al., 2013; Spinoni et al., 2013; Yu et al., 2014a). For example, McEvov et al. (2012) used SPEI as a drought index to monitor conditions over Nevada and Eastern California, proposing that SPEI was a convenient tool to describe the drought in arid regions. Recently, Diasso and Abiodun (2017) investigated the future impacts of global warming and reforestation on drought patterns simulated with the regional climate models over West Africa using the SPEI. Author showed that reforestation over the Savanna could reduce the future warming and increase the precipitation, but the impact of reforestation on the frequency of severe droughts could be doubled."

Page 9, Line 7: "The present study uses SPEI by calculating PET with the Thornthwaite approach, which considers air temperature as a governing feature of PET. However, there are various other methods to calculate PET. For example, the Penman–Montieth method is more physically realistic but requires a diverse input data set (i.e., humidity, radiation coefficient, and wind speed). Van der Schrier et al. (2011) calculated the change in the global Palmer Drought Severity Index (PDSI) using two distinct estimates for PET (e.g., Thornthwaite and Penman–Monteith). The authors found that PSDI based on two PET estimates are identical in terms of trend, average values, and classifying severe wet or dry periods. Conversely, McVicar et al. (2012) suggests that climatic conditions other than temperature that affect PET, may balance temperature rise; therefore, further investigations with multiple approaches could inform future drought characteristics"

In the **Methodology** section, I think the description of the dynamic vegetation module and its functioning should be more detailed. Moreover, I do not understand which parameterization scheme the authors have chosen to represent convection. Related to this point, to ensure the traceability of results, a summary table with all the selected parameterizations could be useful for readers that would like to apply the same modelling set-up over a different region.

>> As per the reviewer's suggestion, we have added one table to show the selected parameterizations for this study.

 Table 2. Model parameterizations used in this study

| Model's feature | Selected schemes        |
|-----------------|-------------------------|
| Boundary layer  | Holtslag PBL            |
|                 | (Holtslag et al., 1990) |

| Cumulus convection      | Emanuel scheme                                  |  |  |  |
|-------------------------|-------------------------------------------------|--|--|--|
|                         | (Emanuel, 1991)                                 |  |  |  |
| Precipitation and cloud | Sub-grid Explicit Moisture Scheme               |  |  |  |
|                         | (Pal et al., 2000)                              |  |  |  |
| Radiation               | Community climate model 3                       |  |  |  |
|                         | (Kiehl et al., 1996)                            |  |  |  |
| Dynamics                | Mesoscale model 5                               |  |  |  |
|                         | (Grell et al., 1994)                            |  |  |  |
| Ocean flux              | Zeng scheme                                     |  |  |  |
|                         | (Zeng et al.,1998)                              |  |  |  |
| Anthropogenic aerosols/ | Tracer model                                    |  |  |  |
| Interactive aerosols    | (Solmon et al., 2006; Zakey et al., 2006, 2008) |  |  |  |
| Land Surface            | Community Land Model 4.5                        |  |  |  |
|                         | (Lawrence et al., 2011; Wang et al., 2016)      |  |  |  |

In terms of run experiments, in my opinion, the study lacks an experiment forced by re-analysis; this extra-experiment would provide a better term of comparison against observations to identify the model biases.

>> This study builds upon the previous studies of Wang et al. (2016) and Erfanian et al. (2016). In particular, Wang et al. (2016) provides extensive model evaluations with the re-analysis data. This point has been clarified in the revised manuscript. However, we agree that the model should be evaluated for capturing the drought characteristics in this study; thus, we have revised section 3.1 to provide the model evaluations with the runs with the ERA-Interim data along with added new figures (Figs 1, 2 and 3).

Page 5, Line 3: "Wang et al. (2016) extensively evaluated the RegCM-CLM-CN-DV model for simulating regional climate and ecosystems in West Africa. The evaluation was performed using the LBCs from the ERA-Interim (1989-2008), and with and without vegetation dynamics. Yu et al. (2016) and Erfanian et al. (2016) also examined the impacts of vegetation dynamics on the climate and ecosystems using multiple LBCs from past and future GCM simulations. Building upon these previous studies, this study focuses on the impacts of vegetation dynamics on the regional drought characteristic (i.e., frequency, duration, and intensity) over the focal regions of the West African domain: the Sahel, the Gulf of Guinea, and the Congo Basin (Fig. 1)."

Page 6, Line 5: "3.1 Model Performance for Present-day Droughts

This section briefly evaluates the model performance with observed climate and vegetation and drought characteristics (Figs 1, 2 and 3). The runs with the ERA-Interim with and without vegetation dynamics for 1989-2008 (Table 1) are briefly presented for the model evaluation. Detailed evaluations of the model performance are documented in Wang et al. (2016). Relative to the observational data from the University of Delaware (UDEL), both EvalSV and EvalDV (Fig. 1) follow the observed spatial patterns of precipitation with slightly underestimating precipitation over the Sahel and overestimating over the Congo Basin. Such dry/wet biases lead to warm/cool biases in air temperature via the reduction/enhancement of evaporative cooling in the Sahel/Congo Basin. In general, the model performs slightly better with SV than with DV in the evaluation runs. But note that DV could eliminate potential consistencies between prescribed vegetation and climate particularly for the future projections.

With the addition of vegetation dynamics, the LAI (Fig. 2) is overestimated in the eastern

parts of Gulf of Guinea and the northern parts of Congo Basin, and it is underestimated in the Sahel (EvalDV-EvalSV). The run without vegetation dynamics (EvalSV) uses the Moderate Resolution Imaging Spectroradiometer (MODIS)-based monthly-varying climatological LAI values. Over the Sahel, the model underestimates the woody plants and grasses with a significant overestimation of bare ground area, which can be attributed to biases in the vegetation dynamics of CLM-CN-DV model as well as the RegCM physical climate, i.e., dry bias (Wang et al., 2016; Erfanian et al., 2016). The dry/wet bias in the atmospheric forcings over the Sahel/Congo Basin contributes to the underestimated/overestimated LAI, which then leads to additional decreases/increases in precipitation for that region.

We also investigated the precipitation surplus/deficit (right column of Fig. 1) that is used for calculating the SPEI values to analyze the drought characteristics. We found that the differences of EvalDV and EvalSV for the precipitation surplus/deficit follow those of the precipitation in these cases. The estimated SPEI over three regions are compared in Fig. 3. While the general cycles of SPEI are limitedly captured in the model, the SPEI differences between UDEL and EvalSV may contribute to the limits of RegCM4. The difference between EvalSV and EvalDV is opposite between the Sahel and other regions, which corresponds to the bases of the precipitation surplus/deficit in Fig. 1. In the Sahel, the more severe and longer droughts are simulated for EvalDV compared with EvalSV. In the Gulf of Guinea and the Congo basin, the opposite was observed."